# Exploring the Associations between Social Media Addiction and Depression: Attentional Bias as a Mediator and Socio-Emotional Competence as a Moderator

**DOI:** 10.3390/ijerph192013496

**Published:** 2022-10-19

**Authors:** Wen Xiao, Jiaxin Peng, Suqun Liao

**Affiliations:** Teacher Education School, Shaoguan University, Shaoguan 512005, China

**Keywords:** mobile social media addiction, depression, anxiety, attentional bias, socio-emotional competencies, adolescents

## Abstract

Social media is used daily by a significant number of young people and can have an important influence on the well-being of its users. The aim of this study was to determine the motives for social media use among Chinese youth and whether social media addiction associates with depression. Another objective was to analyze possible mediating and moderating effects in explaining the association between social media addiction and depression. Participants were 1652 secondary school students (51.5% boys and 48.5% girls) aged 12–18 years in China. The results showed that attention bias mediated the association between social media addiction and depression when adolescents’ socio-emotional competencies were low, but not as strong when their socio-emotional competencies were high. The findings not only provided theoretical support for preventing the negative effects of mobile social media addiction, but could also directly contribute to improving adolescents’ quality of life.

## 1. Introduction

Today’s youth were born into the age of the Internet, and technology and online communication are ubiquitous in the daily lives of many adolescents. According to the latest report of China’s Internet Development in 2019, the Internet access rate among children and adolescents is 93.1% [1]. With the development of digital technology, people now use social media platforms (e.g., Facebook, Twitter, Instagram, Snapchat) to access news, information, and entertainment, post photos, express their opinions, seek company, and maintain networks of friends and family [2]. Social networks have fundamentally changed the interaction and communication patterns of adolescents. While engagement with social media can increase feelings of social connectedness and well-being [3,4], research has largely focused on examining the potential for negative mental health effects from excessive use [5,6]. There is evidence that various online activities, including social media use, have developed a high potential for addiction in adolescents and young adults [7]. Adolescents are also a group in which the emergence of depressive disorders is particularly common [8]. However, it is uncertain how exposure to these technologies might affect the incidence of depressive disorders in this age group. 

Mixed findings are reported regarding the positive and negative psychological outcomes of social media use. Online socializing, under specified conditions, can be beneficial for them. Several studies demonstrate that online social networking may help raise self-esteem and increase sense of belongingness, which can indirectly have a positive influence on psychological well-being [9]. Research indicates that certain kinds of media can actually produce positive or prosocial attitudes among the youth [10] and promote such positive attributes as cooperation and problem-solving skills [11]. According to a recent comprehensive review, playing active video games may even help increase health-promoting physical activities for the participants [12]. With regard to the negative consequences, Internet addiction affects academic performance, social interactions and sleep [13]. Empirical evidence suggests that social media uses were related to negative outcomes such as increased exposure to danger, social isolation, depression, and cyberbullying [9]. For example, Facebook addiction was positively associated with low self-esteem and life satisfaction [14]. Addiction to WeChat (a mobile app most commonly used in China) was negatively correlated with users’ physical, mental, and social health [15]. Social media use was significantly associated with depression [16,17,18]. These studies suggest that depressed participants tend to interact intensively with social media.

Although the significant associations between Internet addiction, well-being, and depression have been examined by extensive empirical literature, the directionality of the causal relationships between addictive social media use and psychosocial problems remain unclear. Both the prior influence of psycho-emotional problems on the development of addiction and the reverse case—the development of problems due to excessive Internet use—are possible. According to the first argument, changes in how adolescents communicate with one another on social media and how much time they spend online account for the increases in psychosocial issues. Evidence supporting this theory revealed that digital social media use causes or significantly worsens a variety of illnesses and issues, including sleep disruption, rising obesity rates, social skill deficits, decreased sexual activity, and an increase in teen mental health issues [19]. Adolescents’ overall screen time (screen time) has increased rates of depression and anxiety [5,20]. A longitudinal study showed that greater positive emotional responses to social media were associated with later depressive symptoms [21]. According to the second argument, teenagers with pre-existing mental health issues may use social media to relieve themselves of stressful symptoms via online connections [22]. For instance, a recent large-scale longitudinal study indicated that early mental health issues in young teenage girls predicted subsequent social media usage but not the other way around [23,24]. Social media use did not predict depressive symptoms over time for males or females. [9,23].

In addition to the relationship of mobile phone use and psychological well-being, research has sought to understand the risk factors associated with social media addiction. Amongst the most widely recognized causes of addiction to social media are low self-esteem, personal dissatisfaction, depression and hyperactivity, and even lack of affection [25]. It was found that personality traits such as neuroticism and extraversion were significantly related to social networking addiction [26]. Teenagers with increased social sensitivity, poor status offline [27], or a history of victimization or bullying [28] are more likely to report negative online experiences. Other relevant factors have been identified that increase the magnitude of this relationship, such as problematic social network use, excessive social comparison, and higher levels of personal engagement [29]. 

The present study aimed to understand the underlying mechanism and examine the possible intermediary processes. In this study, we propose that attentional bias for emotional information mediates the association between addictive mobile social media use and psychological well-being. The cognitive model of depression states that depressed individuals often exhibit attentional and information processing bias, in the form of selective attention to negative aspects of experiences [30]. This process causes the cognitive system to enter a feedback loop that triggers and maintains a depressive episode. Furthermore, unlike depressed individuals, never-depressed individuals withdrew from happy content much more slowly than from neutral or sad content [31]. The authors speculate that this “positive effect” may reflect a protective tendency underlying their ability to keep positive information active [32]. Research on drug and behavioral addiction has shown that addicts tend to process negative content [33], most notably showing an attentional bias toward negative emotions that can exacerbate addictive behavior [34]. Training to shift the attentional focus from negative to positive information can improve the mental health of addicts and effectively reduce addiction [35]. Hu et al. found that college students addicted to mobile phones also exhibited negative emotional processing bias [33]. For these teenagers, using social media could exacerbate mental health issues or cause new ones due to a greater propensity to seek out and repeatedly be exposed to harmful information [27].

The findings from previous studies suggest a relationship between socio-emotional skills and problematic Internet use [36]. Self-control, impulsivity, and social skills were among the cognitive factors that predicted problematic Internet use and fit the concept of socio-emotional competences. Chen et al. found that overall socio-emotional competencies have a significantly negative association with students’ problematic Internet use [36]. The findings from previous studies also suggest a relationship between socio-emotional skills and cyberbullying. A large body of literature confirms that lack of empathy could explain cyberbullying behavior among adolescents [37]. In adolescents, socio-emotional competencies have been shown to predict positive interpersonal relationships and prevent risk behaviors such as technology misuse or violence. Given that adaptive regulation of emotions has a crucial role in successful emotional and cognitive well-being, it was proposed that socio-emotional skill might prevent the negative consequence of social media addiction on mental health.

Moreover, there are significant gender differences in mobile phone addiction. Previous studies have shown that females tend to experience more depressive symptoms than males [38] and are more likely to become addicted to mobile phones because of interpersonal relationships [39]. Females with emotional difficulties, such as subjective unhappiness or depressive symptoms, were found to have a higher risk of developing Internet addiction than males [40]. There were significant correlations between interacting on social media and well-being among females, but not for males [41]. Therefore, the relationship between social media and well-being might differ by gender. This study looks at gender differences in screen-based media interaction and how associations with well-being might differ with gender.

The content presented on social media is colorful and designed to attract attention (e.g., mobile games, short videos, social forums, etc.). Previous results show that social media use and entertainment use have different effects on social media addiction and subjective well-being. Entertainment use tends to lead to social media addiction, while social media use tends to improve subjective well-being [42]. In this study, we also hope to understand which type of mobile use has stronger correlation with psychological well-being. The emotional valence of the socially shared information on social media can drive selection, attention, and sharing behaviors. People who are addicted to social media might become more sensitive and more aware of the emotional stimuli. Therefore, we hypothesized that social media addicts have an attentional bias toward emotion valence information, which may play a mediating role in the relationship between social media addiction and depression. Students with high socio-emotional competencies have been shown to be able to solve social problems, maintain friendships, and regulate emotions and behaviors [43]. Based on these relationships, socio-emotional competencies could be a preventive factor for the association between attention bias and depression. This study included early and late adolescence for whom concerns about social media use and mental health are highest [44]. During the development of self-identity, adolescents are sensitive to reactions from peers and have a strong tendency to compare themselves with others on social media and beyond. 

Based on the empirical evidence and theoretical rationale for the relationships between (a) psychological well-being and social media addiction, (b) psychological well-being and emotional attention bias, and (c) emotional dysregulation and social media addiction, it is plausible to assume that social media addiction promotes the maintenance of emotional attention bias, which in turn leads to depressive disorders. Figure 1 shows the hypothetical model of our study. To the best of our knowledge, this model has not yet been tested. The present study addresses the effects of social media addiction on depression and the underlying mechanisms. We used cross-sectional data collected from a sample of Chinese secondary school students. We aimed to investigate three hypotheses: (a) determining the types of mobile phone use that are related to depression, (b) whether attentional biases act as a mediator between social media addiction and depression in general for all groups, and (c) whether socioemotional competence acts as a moderator that regulates the effects of attentional biases. The mediation hypothesis examined the pathway through which social media dependence correlates with depression in adolescents. The moderation hypothesis examined the condition under which the relationship between attentional biases and depression is strong. The results would help uncover how social media addiction is related to adolescent mental health.

## 2. Materials and Methods

### 2.1. Participants

A cross-sectional study with an unbiased school sample was conducted in a city in southern China. The original sample comprised 1652 adolescents aged 12–18 years. After subtracting those who were unable to complete all sections of the questionnaire (*n* = 630, 38.1%), the sample included 1022 adolescents. Thus, the response rate in this study was 61.9%. The mean age of the final sample (*N* = 1022) was 15.12 ± 1.51 years, with a boy/girl ratio of 526 (51.5%)/496 (48.5%). They were students in a middle school and a vocational school in Shaoguan, Guandong, China. Verbal consent was obtained from the participants before completing the questionnaires at school. They were asked to complete paper self-report questionnaires in a common classroom. The session lasted about 45 min. Demographic and family variables were measured, including age, gender, and frequency of contact between parents and children. Appendix A described the distribution of demographic variables.

### 2.2. Measurements

#### 2.2.1. Attentional Biases

The Attention to Positive and Negative Information Scale (APNIS) [45] measures individual differences in the tendency to attend to, think about, and focus on positive or negative information related to self and others, as well as past and future events. This scale includes 22 questions. Examples of the items were “I pay attention to things that lift me up” and “I tend to focus on the negative things that happen day to day”. Participants were asked to rate the extent to which each statement applied to them on a 5-point Likert scale (from 1 = not at all true of me to 5 = very true of me). In our sample, the Cronbach alpha estimates for attention to positive information (API, 12 items) and attention to negative information (ANI, 10 items) were 0.871 and 0.824, respectively. 

#### 2.2.2. Social-Emotional Competence

Socio-emotional competencies were assessed using the Delaware Social and Emotional Competencies Scale—Student (DSECS-S) [46]. The DSECS-S consists of 12 items, three items in each of the four factors: responsible decision making (e.g., “I blame others when I’m in trouble”), relationship skills (e.g., “I’m good at resolving conflicts with others”), self-management (e.g., “I can control my behaviour”), and social awareness (e.g., “I think about how others feel”). Students were required to respond to each item on a Likert scale ranging from 1 = never to 4 = always. The Cronbach’s alpha for our sample was 0.780.

#### 2.2.3. Anxiety

Anxiety symptom intensity was measured using the 7-item Generalized Anxiety Disorder Test (GAD-7) [47]. Participants were asked to rate their anxiety symptoms over the past 2 weeks on a scale of 0 to 3 ranging from “not at all” to “almost every day”. Example items include: “Feeling nervous, anxious, or on edge” and “Being so restless that it’s hard to sit still”. Scores ranged from 0 to 21, with higher scores indicating higher levels of anxiety. The psychometric properties of GAD-7 with adolescent sample were verified [48]. The Cronbach alpha value for our sample was 0.901. 

#### 2.2.4. Social Media Addiction

The Chinese Social Media Addiction Scale [49] consists of 17 items divided into five dimensions: Salience, Social Gain, Compulsive Use, Withdrawal, and Relapse. Example items are “If I cannot use social media, I feel anxious.” and “I feel more confident through social media communication”. Each item is scored on a 5-point Likert scale (1 = strongly disagree, 5 = strongly agree), with higher scores indicating higher levels of social media addiction. In the present study, the Cronbach’s α value was 0.907.

#### 2.2.5. Mobile Use

We used a Mobile Phone Addiction Type Scale to measure mobile phone use in four categories: Social Media, Gaming, Information Retrieval, and Short Videos. Students were asked to report how many hours per day they typically spend on their mobile phones, including gaming, information retrieval, social media, and short videos, on both school days and non-school days. The average number of hours per day students spent using mobile phones was calculated for each type of screen-based behavior. Response options were based on daily use (0 = never, 1 = not daily, 2 = “≤1 h”, 3 = “2–3 h”, 4 = “3–4 h”, and 5 = “>4 h”).

#### 2.2.6. Depression

Depressive symptoms were assessed using the validated Center for Epidemiological Studies Depression Scale (CES-D) [50]. The scale consists of 20 items that are widely used in studies of the epidemiology of depressive symptomatology in the general population. Participants rate the frequency of occurrence of the symptom on a four-point Likert scale ranging from 0 to 3 to measure the current level of depressive symptomatology, focusing on the affective component, depressed mood. Example items include: “I had trouble keeping my mind on what I was doing.” and “I felt depressed”. The Cronbach’s alpha estimate for our sample was 0.878.

### 2.3. Statistical Analyses

Descriptive statistics, independent samples *t* test, Pearson correlation, and hierarchical multiple regression analyses were performed to detect intercorrelations among core variables using the SPSS 23.0 statistical software package (IBM Corporation, Armonk, NY, USA). The macro PROCESS for SPSS was used to perform the moderated mediation model analysis [51]. The mediation model analysis was performed to test the mediating role of attentional bias. The moderated mediation model analysis was conducted to test the moderating role of socioemotional skills on the indirect effects of attentional bias on depression. We used the bootstrapping method (with 5000 resamples) to generate 95% bias-corrected confidence intervals (CI) of the model effects. CIs that do not contain zero indicate significant effects (α = 0.05).

## 3. Results

### 3.1. Descriptive Statistics

We performed independent samples *t* tests for all other measures to test whether there was any gender difference. Descriptive results are presented in Table 1. The results showed that there were significant gender differences in social media addiction, anxiety, depression, API, ANI, social awareness, and duration of mobile phone use. Compared with boys, girls in our sample scored higher on social media addiction, anxiety, depression, attention bias toward positive and negative information, social awareness, and mobile phone use for social media and watching videos. Boys scored higher than girls on the use of mobile phones for gaming and information retrieval. 

### 3.2. Correlations

The results of the Pearson correlation analysis are shown in Table 2. Social media addiction was negatively correlated with socioemotional competence but positively correlated with anxiety, depression, attentional biases, and mobile device use. In addition, attentional biases, anxiety, and depression were significantly correlated. 

### 3.3. Multiple Regression Analysis between Social Media Addiction and Depression

Hierarchical multiple regression analyses were conducted to test the hypothesis that mobile phone use and addiction may explain variance in depression in addition to variance explained by attentional bias to emotional information. With respect to sex differences, we conducted models separately for males and females. Control variables including age, contact frequency with mother and father, and anxiety score were entered as block 1 items. Then, attentional biases were entered in block 2, and finally, screen time and social media addiction were entered in block 3. 

Multicollinearity tests were calculated and yielded satisfactory indices. Tolerance values were close to 1, and variance inflation factor values were less than 2, indicating that the models were not affected by multicollinearity. The Durbin–Watson statistic was also calculated to test for the absence of autocorrelation in the residuals (prediction error). Values close to 2 were obtained, indicating that there was no autocorrelation in the sample (Hair et al., 2010). Model comparison showed that attentional bias to positive and negative information was significantly associated with depression beyond the controlled variables. The results obtained (Table 3 and Table 4) confirm the relationship with social media and depression. Specifically, mobile games and social media addiction were found to explain the increased depression in boys, and short-form videos were found to explain the increased depression in girls.

### 3.4. Testing for the Moderated Mediation Model of Attention Bias and Socio-Emotional Competences

We conducted a moderated mediation model analysis to examine whether socioemotional competencies could moderate the mediation of attentional biases in the association between social media addiction and depression. We submitted the standardized scores of all the variables to the analysis using model 14 of PROCESS macro for SPSS. The controlled variables including gender, age, and parental contact were entered as covariates. Table 5 shows the results of the moderated mediation model. The direct path coefficient from social media addiction to depression was significant in the absence of mediators (i.e., attentional bias) (*b* = 0.22, *p* < 0.001, 95% CI: 0.16–0.27). In addition, social media addiction significantly predicted attention to positive information (*b* = 0.17, *p* < 0.001, 95% CI: 0.10–0.23) and attention to negative information (*b* = 0.36, *p* < 0.001, 95% CI: 0.30–0.42). When social media addiction and attentional biases were included together as predictors, both attentional biases for positive and negative information significantly predicted depressive symptom severity (*b* = −0.42 and 0.46, respectively, both *p* < 0.001), and the effect of social media addiction on depression was still significant (*b* = 0.22, *p* < 0.001).

The results showed that the cross-term between attention to positive information and socio-emotional competencies was significant for depression (*b* = 0.06, *p* = 0.009, 95% CI: 0.02, 0.11). Therefore, we used conventional procedures to plot simple slopes (see Figure 2) at one standard deviation above and below the mean for socioemotional competencies. The slope of the relationship between attention to positive information and depression was relatively strong (and negative) for adolescents with low socioemotional competencies, whereas the slope was relatively weak for adolescents with high socioemotional competencies. Moreover, the effect of attention to negative information on depression was moderated by socio-emotional skills (*b* = −0.11, *p* < 0.001, 95% CI: −0.16, 0.06). The association between attention to negative information and depression was stronger for individuals with low socio-emotional skills than for individuals with high socio-emotional skills. Figure 3 shows the relationship between attention to negative information and depression for different socio-emotional abilities.

## 4. Discussion

The possibility that adolescents’ use of digital technologies negatively affects psychological well-being is an important question that deserves rigorous empirical testing. Although the relationship between mobile phone addiction and psychological well-being has been examined in many studies, there are still relatively few studies that focus on the mechanism underlying these relationships. The present study examined the factors and mechanisms influencing social media addiction and psychological well-being in adolescents. The results showed that attentional biases toward positive and negative information play a mediating role in the relationship between social media addiction and depression in adolescents. Moreover, socioemotional competencies were able to mitigate the indirect effects of attentional biases, such that the association between attentional biases and depression was stronger when levels of socioemotional competencies were low versus when socioemotional competencies were high. The present study extends previous findings on mobile phone addiction and contributes to the prevention and intervention of mobile social media addiction in adolescents.

Our results show that girls are more likely to be addicted to social media and use mobile phones more than boys. Previous studies have found that females have higher mobile phone addiction than males and that females prefer online social interactions over males [40,52]. Consistent with these findings, girls in our study exhibited relatively higher levels of mobile social media addiction than boys. Because girls have a stronger need for relationships and are more sensitive to rejection, they may avoid problems in interpersonal relationships and be more likely to try to alleviate negative emotions through mobile social interaction. 

Differential uses of mobile phones also have differential effects on students’ psychological subjective well-being. Mobile phone use for short videos significantly predicted depression and anxiety when demographic and other mobile phone use variables were controlled. These results highlight the insights from the use of TikTok, a popular app that encourages young users to watch, create, and comment on “LipSync videos”. Despite the success of this app in terms of user numbers, there are few psychological studies to understand TikTok usage. Our results suggest that certain aspects of TikTok use lead to adverse behavioral effects. 

The novelty of our results is that we investigated the role of attentional biases in the relationship of social media addiction and adolescent depression. The mechanism underlying the correlation between social media addiction and mental health is uncertain. While some studies have confirmed the negative effects of mobile social media addiction on individual development, other studies suggest that it may also have a preventive effect, such as expanding relationships, relaxing, and promoting social support. In terms of attention bias, reliance on social media correlated with attention to both positive and negative information. These results demonstrate the two sides of social media use. One of the main uses of mobile social media among adolescents is to communicate and socially interact with peers through various means, including instant messaging apps (e.g., WhatsApp and social networks). The use of mobile social media offers great opportunities that promote creativity, increase presence and social participation, and provide access to various types of information, including those related to promoting healthy behaviors and habits [53]. However, social media use could also expose adolescents to risks such as cyberbullying, social anxiety, and content that is not age-appropriate or implies negative thoughts. Therefore, social media use may potentially promote both positive and negative information biases. 

In addition, we also uncovered the psychological mechanisms underlying the link between social media addiction and adolescent mental health. We found that attentional biases partially mediated the link between social media addiction and depression. In other words, social media addiction was positively associated with attentional bias for positive and negative information, which in turn correlated with depression. Although previous research has shown that attentional biases are strongly associated with depressive feelings, no previous study has examined the effect of addiction on attentional biases. For the second stage of the mediation model (i.e., the association between attentional biases and depression), our results were consistent with previous studies that have demonstrated the association between attentional biases and depression, specifically. Attention biases to positive and negative information correspond to adolescents’ positive and negative use of social media. Positive use included searching for positive content (i.e., for entertainment, humor, content creation) or for social connection. Negative use included sharing risky behaviors, cyberbullying, and for making self-denigrating comparisons with others. These results provide insights on the effective guidance of social media use in order to prevent the adverse consequence of social media addiction. 

The present study also found that socioemotional competencies significantly moderated the association between attentional biases and depression. The indirect effect of social media addiction on depression through attentional biases was relatively higher for low socio-emotional competence than for high socio-emotional competence. In other words, adolescents with high socioemotional competence were less likely to suffer from depression associated with attentional biases for emotional information and media addiction. As we have noted, socio-emotional skills imply the capacity to establish and maintain healthy friendships and to resolve conflicts [54]. Students with greater relationship skills are more accepted by their classmates and have more friendships [55]. Sociopsychological needs (e.g., need to belong and need for relatedness) are major types of motivations predicting addictive social media use [56]. Findings from a study of United Kingdom adolescents showed that excessive use of social media was associated with more socio-emotional difficulties. High levels of problematic Internet use could be considered as behaviors that compensate for low levels of socio-emotional competencies. Therefore, people with high socio-emotional skills are less likely to be influenced by social media or are better at management of social media use, which prevent them developing negative mental health condition. These results highlight that promoting youth’s overall socio-emotional competencies can be beneficial for preventing the negative consequences brought by social media addiction. 

The present study has some limitations. First, the cross-sectional design cannot clearly confirm that social media addiction causally leads to a change in attentional biases, which in turn increases depression. Future research using a longitudinal design or an experimental study should be carefully considered when evaluating causal influences. Second, the measurement of social media addiction may not be comparable in terms of theoretical basis and scale characteristics. There is still no general consensus on the definition of general Internet addiction, and rigorous and systematic psychometric studies of the measures are lacking. Third, there may be multiple pathways through which social media addiction is associated with depression. In our study, attentional biases only partially mediated the effect of social media addiction on depression. Future research could test other potential mediators that correlate closely with adolescent social media addiction. Fourth, given the weak associations in our study, other macro-level factors in the school environment or micro-level factors in individual psychology might be a better way to help people manage mobile social media addiction and mental health. Future research could explore other protective factors with strong buffering effects to develop more practical interventions. 

## 5. Conclusions

Social media use has a subtle impact on adolescent mental health. Our study found that different types of social media use have different effects on adolescents’ mental well-being. Mobile phone use for short videos had the most negative impact on depression and anxiety. In addition, the study confirmed that social media addiction was correlated with attentional biases toward positive and negative information, depression and anxiety. Specifically, attentional biases for emotional information mediated the relationship between social media addiction and depression, and socioemotional competence moderated this indirect effect. Therefore, colleges and families should pay attention to students’ social media use and guide them to use social media rationally to avoid social media addiction. Social media use, if properly adapted, could also promote healthy behaviors, improve social support, and even become a focal point for information as well as help youths at risk for depression.

## Figures and Tables

**Figure 1 ijerph-19-13496-f001:**
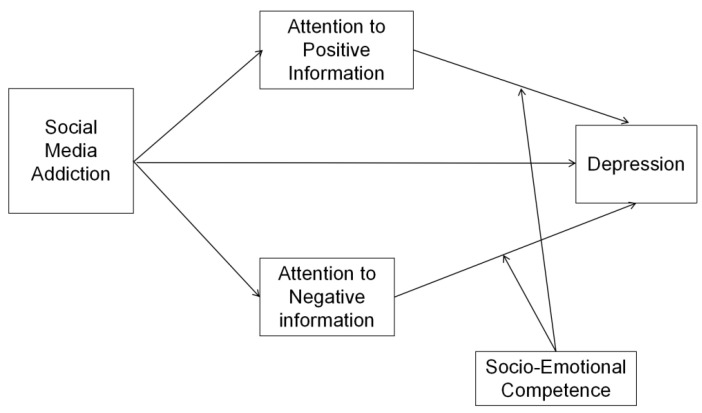
The proposed conceptual scheme.

**Figure 2 ijerph-19-13496-f002:**
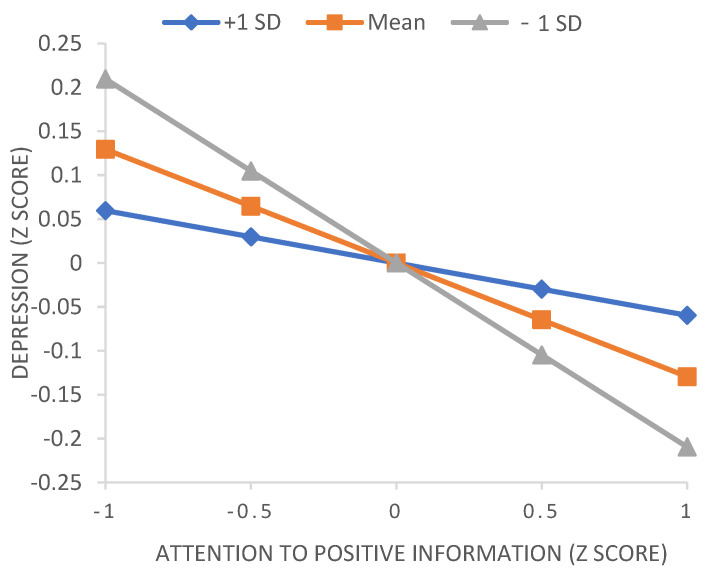
Depression predicted by attention to positive information moderated by socio-emotional skills. +1 SD, one standard deviation above the mean; −1 SD, one standard deviation below the mean.

**Figure 3 ijerph-19-13496-f003:**
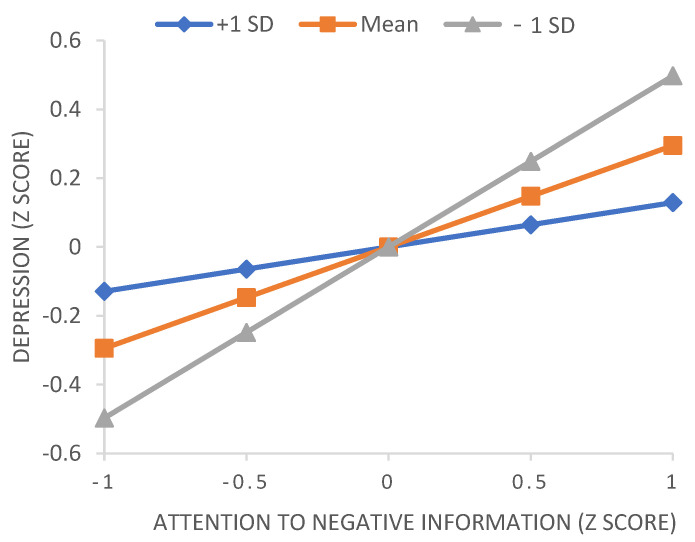
Depression predicted by attention to negative information moderated by socio-emotional skills. +1 SD, one standard deviation above the mean; −1 SD, one standard deviation below the mean.

**Table 1 ijerph-19-13496-t001:** Descriptive statistics for major study variables.

Variables (Score Range)	Whole Sample (*N* = 1022)	Boys (*n* = 526)	Girls (*n* = 496)	*p* (Boys vs. Girls)
Social Media Addiction (1–5)	2.92 (0.72)	2.79 (0.76)	3.05 (0.68)	<0.001
Anxiety (0–21)	4.84 (4.26)	4.11 (3.98)	5.61 (4.40)	<0.001
Depression (0–60)	14.93 (9.63)	13.58 (9.22)	16.35 (9.85)	<0.001
API (1–5)	3.60 (0.66)	3.54 (0.71)	3.66 (0.59)	=0.002
ANI (1–5)	3.26 (0.68)	3.15 (0.71)	3.37 (0.63)	<0.001
Social Emotion Competence (1–4)	3.07 (0.43)	3.05 (0.46)	3.09 (0.39)	0.097
Responsible Decision Making (1–4)	3.16 (0.52)	3.16 (0.54)	3.17 (0.49)	0.923
Relationship Skills (1–4)	3.07 (0.58)	3.06 (0.62)	3.07 (0.55)	0.787
Social Awareness (1–4)	3.05 (0.56)	2.96 (0.58)	3.14 (0.52)	<0.001
Self-Management (1–4)	2.99 (0.56)	3.00 (0.59)	2.98 (0.52)	0.723
Mobile Use				
Gaming (0–5)	2.89 (1.50)	3.10 (1.24)	2.67 (1.70)	<0.001
Information Retrieval (0–5)	2.15 (1.10)	2.25 (1.17)	2.04 (1.02)	0.002
Social Media (0–5)	2.91 (1.34)	2.75 (1.35)	3.08 (1.32)	<0.001
Short Videos (0–5)	3.28 (1.27)	3.19 (1.31)	3.39 (1.21)	0.012

Note: The range of the variables is described in the bracket. Data are presented as mean ± SD. Student’s *t* tests were used. API, attention bias to positive information; ANI, attention bias to negative information.

**Table 2 ijerph-19-13496-t002:** Intercorrelations between the key variables.

	Age	Gender	Addiction	Gaming	Information	Social Media	Videos	Anxiety	Depression	API	ANI
3	**0.18**	**0.18**	\								
4	**0.21**	**−0.14**	**0.16**	\							
5	−0.03	**−0.10**	−0.06	**0.20**	\						
6	**0.31**	**0.13**	**0.36**	**0.30**	**0.16**	\					
7	**0.32**	0.08	**0.27**	**0.32**	**0.13**	**0.52**	\				
8	**−0.15**	**0.18**	**0.32**	0.01	−0.06	0.08	**0.10**	\			
9	−0.05	**0.14**	**0.32**	**0.09**	−0.03	**0.12**	**0.17**	**0.71**	\		
10	**−0.09**	**0.10**	**0.15**	−0.03	0.08	0.07	0.02	−0.09	**−0.22**	\	
11	**−0.10**	**0.16**	**0.34**	0.03	0.04	**0.10**	0.07	**0.37**	**0.35**	**0.45**	\
12	−0.04	0.05	**−0.10**	**−0.11**	**0.09**	0.02	−0.03	**−0.22**	**−0.28**	**0.42**	0.09

Note: 1. Age; 2. Gender (1 = boy, 2 = girl); 3. Social media addiction; 4. Screen time for mobile games; 5. Screen time for information retrieval; 6. Screen time for social media; 7. Screen time for short videos; 8. GAD-7; 9. CES_D; 10. Attention to positive information; 11. Attention to negative information; 12. Socio-emotional competences. Bold values denote statistical significance at the *p* < 0.001.

**Table 3 ijerph-19-13496-t003:** Hierarchical multiple regression analyses for depression in girls (*n* = 496).

Variables	R^2^	F(df)	*p*	Standardized β	*p*
Step 1	0.576	166.86 (491)	<0.001		
Age				−0.084	0.007
Contact frequency with father				0.092	0.001
Contact frequency with mother				0.070	0.015
Anxiety				0.575	<0.001
Step 2	0.655	154.50 (489)	<0.001		
API				−0.285	<0.001
ANI				0.255	<0.001
*p* for F change Model 1 versus Model 2	0.079	55.59 (489)	<0.001		
Step 3	0.662	89.20 (484)	<0.001		
Mobile Games				0.000	0.998
Information Acquisition				−0.008	0.761
Social Media				0.036	0.284
Short-form Videos				0.095	0.002
Social Media Addiction				0.049	0.128
*p* for F change Model 2 versus Model 3	0.015	4.40 (484)	0.001		

**Table 4 ijerph-19-13496-t004:** Hierarchical multiple regression analyses for depression in boys (*n* = 526).

Variables	R^2^	F(df)	*p*	Standardized β	*p*
Step 1	0.460	111.16 (521)	<0.001		
Age				−0.010	0.789
Contact frequency with father				−0.004	0.904
Contact frequency with mother				0.110	0.002
Anxiety				0.541	<0.001
Step 2	0.507	88.81 (519)	<0.001		
API				−0.264	<0.001
ANI				0.186	<0.001
*p* for F change Model 1 versus Model 2	0.046	55.59 (519)	<0.001		
Step 3	0.529	52.39 (514)	<0.001		
Mobile Games				0.101	0.008
Information Acquisition				0.033	0.321
Social Media				−0.082	0.033
Short-form Videos				0.040	0.316
Social Media Addiction				0.107	0.003
*p* for F change Model 2 versus Model 3	0.022	4.40 (51)	0.001		

**Table 5 ijerph-19-13496-t005:** Moderated mediation analysis.

Regression Model Predicting Depression	β	se	*t*	95% CI
Constant	−0.02	0.03	−0.63	[−0.07, 0.03]
Gender	0.07	0.03	2.90 **	[0.02, 0.12]
Age	−0.10	0.03	−3.84 **	[−0.16, −0.05]
Contact frequency with father	0.04	0.03	1.59	[−0.01, 0.10]
Contact frequency with mother	0.07	0.03	2.71 **	[0.02, 0.13]
Social media addiction	0.22	0.03	7.80 ***	[0.16, 0.27]
Attention to positive information	−0.42	0.03	−13.65 ***	[−0.48, −0.36]
Attention to negative information	0.46	0.03	15.57 ***	[0.40, 0.52]
Socio-emotional competencies	−0.13	0.03	−4.42 ***	[−0.18, −0.07]
Attention to positive × Socio-emotional competencies	0.06	0.02	2.60 **	[0.02, 0.11]
Attention to negative × Socio-emotional competencies	−0.10	0.03	−4.06 **	[−0.15, −0.05]
Conditional effect analysis of API at values of the moderator (Socio-emotional competencies)				
−1 SD	−0.48	0.04	−11.92 ***	[−0.56, −0.40]
mean	−0.42	0.03	−13.65 ***	[−0.48, −0.36]
1 SD	−0.36	0.04	−9.62 ***	[−0.43, −0.28]
Conditional effect analysis of ANI at values of the moderator (Socio-emotional competencies)				
−1 SD	0.57	0.04	13.64 ***	[0.48, 0.65]
mean	0.46	0.03	15.57 ***	[0.40, 0.52]
1 SD	0.36	0.04	9.90 ***	[0.29, 0.43]

Note. *N* = 1022. Standardized regression coefficients are reported. Bootstrap sample size = 5000. LL, low limit; CI, confidence interval; UL, upper limit.** *p* < 0.01. *** *p* < 0.001.

## Data Availability

The datasets generated for this study are available on request to the corresponding author.

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
