# Peer review of "Exploring the Associations between Social Media Addiction and Depression: Attentional Bias as a Mediator and Socio-Emotional Competence as a Moderator"

_ijerph, 2022, doi:10.3390/ijerph192013496_

Round 1

Reviewer 1 Report

The authors aimed at the associations between social media addiction and psychological well-being, including attentional bias as a mediator and socio-emotional competence as a moderator in the age group 11-19.

Previous research concerning the topic of the article has been sufficiently presented. The results of the study are described in detail in the text and presented in tables. The discussion is well written.

However, I would like to draw the authors' attention to several important issues:

-         In the introduction part, I recommend to the authors add information about other internet addiction triggers. Also important are parental control or personality factors. It is also worth mentioning the "challenges" appearing on the Internet in recent years, in which young people participate, which often pose a threat to life.

-         Authors state (lines 32-33) that “While engagement with social media can increase feelings of social connectedness and well-being”. I suggest expanding the paragraph to include more detailed positive effects of using the Internet. As a result, the reader will receive more information about both positive and negative effects.

-         In lines 92-94, 119-121, and 124-132 the authors present the aims of their research. I would recommend the authors put this information together in one paragraph. This will make the text clear.

-         I want to point out that the authors recruited a wide range of age participants. It includes the period of early and late adolescence. I would expect the authors to justify that such a group selection does not interfere with the results, or to include this issue in the limitations of the research.

-         I recommend the authors graphically present the number of participants about their age.

-         I suggest moving part 2.2.8 to the 2.2.1 Participants, as well as adding tables to the appendix with the descriptive statistics

-         The SWLS and GAD-7 questionnaires are designed for adults. I would expect the authors to justify the choice of the above-mentioned methods.

-         I propose to add information (2.2.6. Mobile use) regarding the number of hours participants spend on mobile phones. Was it a subjective assessment of the participants or an objective assessment based on the calculations of the phone application?

-         Lines 214-218 should be included in the group description.

-         In Table 2, the authors included gender. Gender is a nominal variable. If the authors decide to leave the gender variable in the table, I would ask for a statistical interpretation of this result.

Reviewer 2 Report

The manuscript entitled „Exploring the associations between social media addiction and psychological well-being: Attentional bias as a mediator and socio-emotional competence as a moderator” presents a cross-sectional study on the associations between social media addiction and depression in adolescents. The Authors examined whether this association is mediated by attentional bias, and moderated by socio-emotional competencies. The goal of the study is important from the public health perspective. The Authors conducted their study in large sample which warrants an appropriate power. However, I have some suggestions which should be solved prior the publication of the manuscript.

#1. The most important drawback of the study is its cross-sectional design. The Authors are aware of this limitation. However, throughout the manuscript it is present a causal language describing the analysis or results. It should be corrected. The Authors could also improve their theory by referring to the findings from longitudinal studies that showed causal associations between sm addiction and depression, eg.

Jacqueline Nesi, W. Andrew Rothenberg, Alexandra H. Bettis, Maya Massing-Schaffer, Kara A. Fox, Eva H. Telzer, Kristen A. Lindquist & Mitchell J. Prinstein (2021): Emotional Responses to Social Media Experiences Among Adolescents: Longitudinal Associations with Depressive Symptoms, Journal of Clinical Child & Adolescent Psychology, DOI: 10.1080/15374416.2021.1955370

But also discuss those which did not support this direction of associations:

Puukko et al., 2020; Int. J. Environ. Res. Public Health 2020, 17, 5921; doi:10.3390/ijerph17165921

Heffer, T., Good, M., Daly, O., MacDonell, E., & Willoughby, T. (2019). The Longitudinal Association Between Social-Media Use and Depressive Symptoms Among Adolescents and Young Adults: An Empirical Reply to Twenge et al. (2018). Clinical Psychological Science, 216770261881272. doi:10.1177/2167702618812727

Better support for the model tested in the manuscript is needed. Moreover, please indicate why the model contains only depression as DV, while anxiety and well-being were also assessed as DV?

#2. Please include the statistics for a simple slope analysis of moderated mediation. In the current form we only know that the interaction was significant. However, in order to interpret the results the betas for simple slopes should be also provided with t-test results.

#3. In Table 2, in my opinion, is better to bold significant correlations compared to mark the non-significant correlations.

#4. Gender differences should be better described in the introduction and possibly tested in the regression model. Some studies (Heffer et al., 2019) showed that the associations between sm addiction and depression were present among girls, but non-significant among boys. Thus, the Authors could tested whether the regression models are similar in both gender.

#5. How did the Authors control the analyses for family and personal variables which were collected during the study?

#6. In the discussion the Authors refer to problems in emotion regulation trying to explain their results regarding socio-emotional competencies. However, please remember that dysregulation was not assessed in the measure which was used to measure socio-emotional competencies in the present study. The Authors used a measure which covers decision making, relationship skills, self-management and social awareness. Moreover, the measure was short (12 items). Please, provide also exemplary items for all scales used in the study and some references about their validity.

#7. Some grammar and spelling errors are present, e.g. Differential use of mobile phones also has differential effects on students' psycholog- 325 ical subjective well-being; Our study found 389 that different types of social media use have different effects on adolescents' mental well-being. mobile phone

Round 2

Reviewer 1 Report

Thank you for following my recommendations. The paper has been substantially improved and I recommended the article for publication in the journal.

Author Response

Thank you very much for your previous comments that helped us improve this manuscript.
